# Enhanced Therapeutic Effect of Optimized Melittin-dKLA, a Peptide Agent Targeting M2-like Tumor-Associated Macrophages in Triple-Negative Breast Cancer

**DOI:** 10.3390/ijms232415751

**Published:** 2022-12-12

**Authors:** Soyoung Kim, Ilseob Choi, Ik-Hwan Han, Hyunsu Bae

**Affiliations:** 1Department of Physiology, College of Korean Medicine, Kyung Hee University, 26 Kyungheedae-ro, Dongdaemun-gu, Seoul 02447, Republic of Korea; 2Department of Science in Korean Medicine, College of Korean Medicine, Kyung Hee University, 26 Kyungheedae-ro, Dongdaemun-gu, Seoul 02447, Republic of Korea; 3Convergence Innovation Support Center, Gangwon Technopark, Chuncheon-si 24341, Republic of Korea

**Keywords:** triple-negative breast cancer, melittin-dKLA, PEG-melittin-dKLA 8-26, tumor microenvironment, tumor-associated macrophages, cancer immunotherapy

## Abstract

Triple-negative breast cancer (TNBC) is characterized by a high possibility of metastasis. M2-like tumor-associated macrophages (TAMs) are the main components of the tumor microenvironment (TME) and play a key role in TNBC metastasis. Therefore, TAMs may be a potential target for reducing TNBC metastasis. Melittin-dKLA, a peptide composed of fused melittin and pro-apoptotic peptide d(KLAKLAK)2 (dKLA), showed a potent therapeutic effect against cancers by depleting TAMs. However, melittin has a strong adverse hemolytic effect. Hence, we attempted to improve the therapeutic potential of melittin-dKLA by reducing toxicity and increasing stability. Nine truncated melittin fragments were synthesized and examined. Of the nine peptides, the melittin-dKLA8-26 showed the best binding properties to M2 macrophages and discriminated M0/M1/M2. All fragments, except melittin, lost their hemolytic effects. To increase the stability of the peptide, melittin-dKLA8-26 fragment was conjugated with PEGylation at the amino terminus and was named PEG-melittin-dKLA8-26. This final drug candidate was assessed in vivo in a murine TNBC model and showed superior effects on tumor growth, survival rates, and lung metastasis compared with the previously used melittin-dKLA. Taken together, our study showed that the novel PEG-melittin-dKLA8-26 possesses potential as a new drug for treating TNBC and TNBC-mediated metastasis by targeting TAMs.

## 1. Introduction

Triple-negative breast cancer (TNBC) is a subtype of breast cancer that accounts for 10–15% of breast cancers, characterized by a lack of estrogen receptor (ER), progesterone receptor (PR), and an expression of low levels of human epidermal growth factor 2 (HER2) [1,2]. Compared to other breast cancer subtypes, TNBC patients are prone to be short-lived. TNBC is highly invasive, and about 46% of TNBC patients show bone, brain, liver, and lung metastasis [3,4,5]. The tumor microenvironment (TME) plays a key role in tumor progression, metastasis, and poor progression of TNBC [6,7,8,9,10]. Hence, an understanding of the interaction between TNBC cells and the TME is necessary to identify new therapies for TNBC.

Tumor-associated macrophages (TAMs) are immune cells abundant in the TME [11,12,13]. They can be divided into classically activated or antitumoral M1-like and alternatively activated or pro-tumoral M2-like TAMs. M2-like TAMs contribute to tumor growth, angiogenesis, migration and invasion, and metastasis [14,15]. It has been reported that the infiltration of TAMs induces remote metastasis and causes lower rates of survival in cancer patients [16]. Recently, Chen et al. reported that M2-like macrophages promote epithelial–mesenchymal transition (EMT) and cancer stem cell (CSC) properties in TNBC [17]. Therefore, targeting M2 macrophages is necessary for treatment.

Melittin is a major component of honeybee venom and consists of 26 amino acids. Melittin is an antibacterial, antifungal, and antitumor agent [18,19]. When melittin attaches to the lipid membrane [19], membrane fluctuation is vigorously induced, and melittin interferes with the deformed membrane [20]. As asymmetry between the two lipid layers occurs, the force of melittin entering the membrane decreases, and a pore is formed temporarily. This series of processes induces cell death [21]. Melittin has been investigated for cancer treatment because it is cytotoxic to diverse tumor cell lines and inhibits cell growth [22,23,24]. On the other hand, it was reported that a low dose of melittin inhibits pore formation [25,26]. Lee et al. found that in the range of concentration used in vitro, melittin selectively decreased M2 macrophages and did not cause cell lysis [27]. (KLAKLAK)2 (KLA) is an antibacterial peptide that binds to negatively charged bacterial membranes and disrupts them [28,29]. Note that the bacterial outer membrane is negatively charged, whereas the outer membrane of mammalian cell cells is neutral. It has been argued that KLA preferentially binds to bacteria compared to eukaryotic cells because antimicrobial peptides are positively charged [28]. However, when KLA is internalized by eukaryotic cells, it can disrupt the mitochondrial membrane of eukaryotic cells, which is similar to the bacterial membrane [30]. In this respect, KLA has been widely used as a payload in drug conjugate design. In previous studies, a peptide drug was designed using a GGGGS linker by conjugating melittin and the pro-apoptotic peptide d(KLAKLAK)2 (KLA), which is an all-D enantiomer, to avoid degradation by proteases. This peptide drug (so-called melittin-dKLA) has been shown to reduce tumor growth by selectively depleting M2-like TAMs in lung cancer and melanoma model [27,31,32]. Despite its effectiveness, melittin causes toxicity in red blood cells, even at low concentrations, to induce strong hemolysis and secrete hemoglobin into the cell to cause toxicity [33,34,35,36].

This study aimed to determine whether melittin-dKLA modification enables the targeting of M2-like TAMs with weak hemolytic activity and superb effects on the inhibition of tumor growth and progression in TNBC.

## 2. Results

### 2.1. Polarization of Macrophages in THP-1-Derived Macrophages

To identify the differentiation of each macrophage phenotype, THP-1 cells were treated with lipopolysaccharides (LPS) and interferon (IFN)-γ for M1 and interleukin (IL)-13 and IL-4 for M2. Note that IFN-γ is a type II interferon secreted by T helper type 1 (Th1) cells and plays an important role in innate and adaptive immunity. IFN-γ is a known important activator of M1 macrophages. IL-13 and IL-4 are cytokines secreted by Th2 cells. In macrophages, IL-4 and IL-13 are known to induce the activation of M2 macrophages. The cells were identified by assessing the morphology or the expression of M1 or M2 macrophage markers. As shown in Figure 1A, THP-1 cells were round and nonadherent, whereas phorbol 12-myristate 13-acetate (PMA)-treated THP-1 cells were adherent. M1 macrophages showed roundish cell bodies and elongated cytoplasmic extensions, while M2 macrophages showed a roundish cell body and shorter and thicker cytoplasmic extensions (Figure 1A). As shown in Figure 1B,C, the expression of CCR7, TNF-α, and CD86, which are M1 macrophage markers, increased in M1 macrophages. Furthermore, M2 macrophage markers such as Arginase-1, CD206, and CD163, were increased in M2 macrophages. Hence, these polarized cells were used in subsequent experiments.

### 2.2. Melittin Fragments Has Fewer Hemolytic Effects than Melittin

Melittin has been well-documented to have strong hemolytic effects. To diminish such adverse effects, several truncated melittin were synthesized and analyzed in vitro (Figure 2A). Mouse blood cells were treated with varying concentrations of peptides for 4 h, and the optical density was measured using a microplate reader. Hemolytic effects of all melittin fragments were dramatically decreased, while melittin treatment at a concentration above 5 μM showed hemolysis from the point of treatment (Figure 2B). This result indicates that melittin truncation resulted in significantly lower hemolytic effects.

### 2.3. Binding of Melittin Fragments to the THP-1-Derived M2 Macrophages

To address the binding ability of truncated melittin peptides to M2 macrophages, fluorescein isothiocyanate (FITC)-conjugated peptides were synthesized and incubated with M2 macrophages, and the percentage of FITC-positive cells was measured using flow cytometry. Melittin was the peptide with the highest binding affinity to M2 macrophages, followed by the melittin fragment containing amino acid residues 8-26 (melittin 8-26) (Figure 3A). Peptides other than melittin 8-26 showed negligible binding affinities to M2 macrophages (Figure 3A). Next, to confirm and compare the binding capabilities of melittin and melittin 8-26 to different macrophage lineages, including M0, M1, and M2, FITC-conjugated peptides were tested on each macrophage phenotype, and their binding capabilities were measured by flow cytometry. Melittin bound to M0, M1, and M2 at rates of 21%, 17%, and 46%, respectively, while melittin 8-26 bound to M0, M1, and M2, at rates of 26%, 14%, and 34% respectively (Figure 3B). These results indicate that melittin 8-26, which has a smaller hemolytic effect than melittin, preferentially binds to M2 macrophages such as melittin.

### 2.4. Characterization of the Optimized Hybrid Peptide Drug, PEG-Melittin-dKLA 8-26

The optimized peptide fragment melittin 8-26 was used as a vehicle for targeting M2 macrophages, and the pro-apoptotic peptide d(KLAKLAK)2 (dKLA) was conjugated to the C-terminus of the peptide to test its cytotoxic effects on M2 macrophages. The half-maximal (50%) inhibitory concentration (IC50) of melittin-dKLA was evaluated as 1.229, 0.737, and 0.415 μM in M0, M1, and M2, respectively, while the IC50 of melittin-dKLA 8-26 was measured as 4.125, 2.185, and 1.454 μM in M0, M1, and M2 macrophages, respectively (Figure 4A). These results suggest that even though the general binding efficacy was decreased, the optimized peptide melittin-dKLA 8-26 possessed selective cytotoxic effects between M0, M1, and M2 macrophages, similar to the original melittin-dKLA. Next, to ascertain the loss of hemolytic effects of the optimized peptide, melittin-dKLA 8-26 was tested using an in vitro hemolysis assay. Hemolysis was observed at a concentration of 5 μM melittin-dKLA. However, there were no hemolytic effects of melittin-dKLA 8-26 at the tested concentration of up to 40 μM (Figure 4B). These data indicate that compared to melittin-dKLA, melittin-dKLA 8-26 selectively targets M2 macrophages without serious hemolysis. Additionally, peptide drugs have the disadvantage of being easily decomposed in the body. Peptide drugs are stabilized using various methods, such as PEGylation. In this study, melittin-dKLA 8-26 was PEGylated at the amino terminus to increase its stability in the body. This final optimized peptide was named PEG-melittin-dKLA 8-26 and was subjected to a stability assay in human serum. The stability of melittin-dKLA 8-26 was further increased on PEGylation (Figure 4C). Thus, our findings suggest that melittin-dKLA 8-26 targets M2 macrophages with a low hemolytic effect, as well as shows improved stability through PEGylation compared with melittin-dKLA, previously described as a potent agent for cancer therapy.

### 2.5. Inhibition of Tumor Growth and Increased Survival Rates by PEG-Melittin-dKLA 8-26 in 4T1 TNBC Mouse Model

To evaluate the efficacy of PEGylated (PEG)-melittin-dKLA 8-26 in a mouse triple-negative breast cancer (TNBC) model, BALB/c mice were injected with 4T1 cells and melittin-dKLA or PEG-melittin-dKLA 8-26 was administered (Figure 5A). Treatment with melittin-dKLA and PEG-melittin-dKLA 8-26 significantly decreased tumor growth (Figure 5B,C). Survival analysis showed that the median survival periods were 31, 33, and 35 days in the phosphate-buffered saline (PBS)-treated, melittin-dKLA-treated, and PEG-melittin-dKLA 8-26-treated groups, respectively. The results showed that survival rates significantly increased in the PEG-melittin-dKLA 8-26 treatment than in the PBS- or melittin-dKLA-treated groups (Figure 5D), suggesting that PEG-melittin-dKLA 8-26 seems to be an effective peptide for improving tumor growth and survival rates outcomes.

### 2.6. Enhanced Suppression of Lung Metastasis by PEG-Melittin-dKLA 8-26 in the TNBC Mouse Model

To test the effects of peptides on lung metastasis in the 4T1 TNBC model, lung tissues were stained, and lung nodules were examined (Figure 6A). PEG-melittin-dKLA 8-26 showed more significant effects in decreasing lung nodule area and the number of lung nodules than melittin-dKLA (Figure 6B–D). These results suggest that PEG-melittin-dKLA 826 is more effective than melittin-dKLA in suppressing lung metastasis in the TNBC model.

### 2.7. Reduction of M2-like TAMs by PEG-Melittin-dKLA 8-26 in the 4T1 TNBC Mouse Model

To determine and compare whether M2-like tumor-associated macrophages (TAMs) were reduced in tumor tissue, tumor tissue was collected, RNA was isolated, and the expression of M2 macrophage markers was measured by RT-PCR. In the PEG-melittin-dKLA 8-26-treated group, the expression of the transforming growth factor (TGF)-β and arginase 1, which are M2 macrophage markers, was significantly reduced. In addition, the expression of matrix metalloproteinase 9 (MMP9) and vascular endothelial growth factor (VEGF) was significantly reduced upon PEG-melittin-dKLA 8-26 treatment. These data suggest that PEG-melittin-dKLA 8-26 further reduced M2-like TAMs compared with melittin-dKLA.

## 3. Discussion

Our study demonstrated that the newly optimized PEG-melittin-dKLA 8-26 targets M2-like tumor-associated macrophages (TAMs) with weak hemolysis and suppresses lung metastasis in a triple-negative breast cancer model more effectively than melittin-dKLA, which has been previously reported as a potent agent for cancer treatment.

Breast cancer is a heterogeneous malignancy commonly diagnosed in women. Triple-negative breast cancer (TNBC) is a breast cancer subtype that is negative for estrogen and progesterone receptors and HER2. Because TNBC lacks therapeutic targets, it is impossible to use drugs such as HER2-targeted drugs (herceptin); thus, it is necessary to develop new treatments and targets for TNBC. Owing to its rapid growth, TNBC spreads to remote organs and is likely to recur compared with other subtypes. It is considered an aggressive breast cancer subtype [37]. The above-mentioned characteristics explain why the development of drugs against metastatic TNBC is necessary. Tumor-infiltrating lymphocytes and tumor-associated macrophages are more abundant in the tumor microenvironment (TME) of TNBC than in other breast cancer subtypes [3,38].

TAMs are tumor-infiltrating macrophages that are the major component of the TME and are formed in solid tumors [39]. In breast cancer, TAMs account for 50% of the total number of cells in the TME [40]. Macrophages are classified as M1 and M2. M1 macrophages are pro-inflammatory, antitumoral macrophages that are induced by the tumor necrosis factor (TNF)-α, interferon (IFN)-γ, and lipopolysaccharide (LPS) and express CD86 and inducible nitric oxide synthase (iNOS). M2 macrophages are anti-inflammatory, pro-tumoral macrophages that are induced by interleukin (IL)-4 and IL-13 and express CD163, CD206, and Arginase-1 [41]. M2-like TAMs have been reported to contribute to invasive breast cancer formation in a transplantable p53-null mouse model [40]. Furthermore, the colony stimulating factor (CSF)-1 inhibitor reduces tumor growth and progression and prolongs survival by suppressing TAM infiltration in mouse mammary cancers [42]. Local and systemic levels of matrix metalloproteinase 9 (MMP9), vascular endothelial growth factor (VEGF), chitinase-3-like protein 1, and lipocalin-2 induced by TAMs result in metastasis in TNBC [41,43,44,45]. TAMs suppress T cell and natural killer cell activation by expressing or releasing immunosuppressive factors such as arginase 1 (ARG1), indoleamine 2,3-dioxygenase (IDO), IL-10, programmed death ligand 1 (PD-L1), and transforming growth factor (TGF)-β [46]. In addition, TAMs recruit immune-suppressive regulatory T cells, which suppress IFN-γ secretion by CD8+ T cells [47]. Because TAMs contribute to tumor development and progression and a high density of TAMs results in poor prognosis in various cancers, it is necessary to develop anticancer drugs targeting M2-like TAMs [48,49].

Melittin is an amphipathic peptide with 26 amino acid residues and is a major component of honeybee venom [19]. Lee et al. demonstrated that melittin preferentially bound to CD11b^+^ cells rather than to CD4^+^ cells and CD8^+^ cells in spleen cells. In CD11b+ cells, melittin preferentially bound to F4/80+ macrophages rather than CD11c+ dendritic cells or Gr-1+ neutrophils. Melittin is also preferentially bound to CD206+ M2 macrophages rather than CD86+ M1 macrophages. These results were not associated with the phagocytic activity of macrophages because the inhibition of macrophage actin polymerization does not affect the melittin binding to macrophages. In addition, the phagocytosis capacity of M1 macrophages with higher phagocytic potential was not affected by melittin [27]. Therefore, we assumed that melittin is preferentially bound to M2 via receptor binding. In subsequent studies, Lee et al. investigated the therapeutic effect of a designer peptide that conjugates melittin and the all-D enantiomer form of (KLAKLAK)2 (dKLA) to avoid degradation by proteases in a cancer model and found antitumor effects of melittin-dKLA in a lung cancer [31]. The melittin-dKLA peptide selectively binds to F4/80^+^ CD206^+^ M2-like TAMs and suppresses tumor growth and progression in mouse melanoma and TNBC models [32,50]. Melittin can cause severe toxic reactions, such as hemolysis, the most serious side effect of melittin, when injected intravenously [18,51,52,53]. Hence, it is necessary to decrease the hemolytic effects of melittin, to make it more effective and safer for cancer treatment.

In this study, we identified nine truncated peptides of melittin that reduced hemolysis (Figure 2). In addition, similar to melittin, melittin 8-26 corresponds to amino acid sequences 8 to 26 of melittin and preferentially binds to M2 macrophages compared with other macrophage phenotypes (Figure 3). Melittin-dKLA has been reported to selectively induce apoptosis in M2 macrophages by activating caspase 3 [32]. In this study, we tested the macrophage cytotoxicity and hemolysis of melittin-dKLA or melittin-dKLA 8-26. We found that like melittin-dKLA, melittin-dKLA 8-26 targeted M2 macrophages better than M0 and M1 macrophages. However, melittin-dKLA 8-26 showed less hemolysis than melittin-dKLA (Figure 4A,B).

Polyethylene glycol has been widely utilized as a polymer in biopharmaceutics. PEGylation is a technique that combines drugs and polyethylene glycol for effective delivery. It improves delivery by improving the penetration of biological barriers, such as extracellular matrix of tissues, cellular barriers, and biological fluids [54,55,56,57]. Recently, Seo et al. demonstrated that PEGylation contributes to better stabilization of peptides. The authors reported greater metabolic stability of PEGylated dimerized translationally controlled tumor protein blocking peptide (dTBP2) than that of natural dTBP2 in human and mouse liver microsomes [58]. Currently, several PEGylated peptide drugs have been FDA-approved and are candidates for clinical trials [59,60,61,62,63]. As expected, when melittin-dKLA 8-26 was PEGylated, the stability in serum increased compared with melittin-dKLA (Figure 4C). Based on the in vitro results, it was necessary to compare the effects of melittin-dKLA and PEG-melittin-dKLA 8-26 in vivo. Although there was no improvement effect on tumor growth, it was found that PEG-melittin-dKLA 8-26 had a superior effect on survival rate and the inhibition of lung metastasis compared to melittin-dKLA (Figure 5 and Figure 6). Additionally, PEG-melittin-dKLA 8-26 treatment significantly reduced various M2-related markers relative to melittin-dKLA, suggesting that PEGylation might help to increase peptide efficacy in vivo.

Breast cancer metastasizes to distant organs, including the bone, brain, liver, lung, and distant lymph nodes [1,5]. Several trials have attempted to treat metastatic TNBC using various drugs. Chemotherapeutic drugs such as anthracyclines, taxanes, gemcitabine, and capecitabine are commonly used alone or in combination for the treatment of TNBC. Additional drugs, such as immune checkpoint inhibitors, antibody–drug conjugates, and immunotherapy drugs, have been developed to effectively treat TNBC [64]. For instance, in the KEYNOTE-355 trial, the treatment of TNBC with the combined administration of pembrolizumab programmed cell death protein (PD)-1 inhibitor and chemotherapy drugs (paclitaxel, nab-paclitaxel, or carboplatin/gemcitabine) was investigated to identify improved progression-free survival in patients with locally recurrent inoperable or metastatic TNBC that show tumor expression of PD-L1 [65]. LY2109761, an inhibitor of TGF-β receptor I/II, has also been shown to reduce the migration and metastasis of TNBC cells [66]. Treatment with selumetinib, an MEK inhibitor, induced mesenchymal to epithelial transition, reduced the cancer stem cell population in vitro, and inhibited lung metastasis in the MDA-MB-231 xenograft model [67]. Our previous study demonstrated that melittin-dKLA effectively suppressed TNBC lung metastasis through the elimination of M2-like TAMs, accompanied by a reduction in the expression of tumor-development-related genes, such as CD44, CC motif chemokine ligand (CCL)22, hypoxia-inducible factor (HIF)-1α, and MMP9 [50]. Melittin-dKLA effectively inhibited lung metastasis; however, PEG-melittin-dKLA 8-26 was more effective in suppressing lung metastasis (Figure 6). In summary, PEG-melittin-dKLA 8-26 is prone to reduce lung metastasis compared with melittin-dKLA by reducing the expression of M2-like TAMs, which are known to contribute to metastasis. We found that the PEG-melittin-dKLA 8-26 treatment significantly reduced various markers induced by M2-like TAMs (Figure 7). However, despite this result, an additional study about the specific effect of PEG-melittin dKLA 8-26 on M2 macrophages and the specific binding capability of PEG-melittin-dKLA 8-26 to the other immune cells needs to be carried out.

A previous study and current study have led us to assume that M2 macrophages may have specific binding molecules for melittin. However, the specific binding molecules of melittin or melittin 8-26 on M2 macrophages are veiled. Hence, using LC/MS/MS proteomics, or CRISPR/cas9 library screening, the investigation of specific binding molecules for melittin and melittin 8-26 on M2 macrophages is currently underway. Immune checkpoint inhibition is a mechanism that inhibits the expression of immune checkpoint proteins, such as PD-1/PD-L1 and cytotoxic T-lymphocyte antigen (CTLA)-4/B7-1/B7-2, expressed on T cells or cancer cells. As immune checkpoints induce immune tolerance of T cells and help cancer cells to evade immune responses, this mechanism has been investigated for cancer immunotherapy [68]. Recently, it was reported that higher levels of PD-L1 are present in metastatic TNBC than in other subtypes, suggesting that immunotherapy is a promising approach for TNBC treatment [3]. A previous study found that TAM depletion with a colony-stimulating factor-1 receptor (CSF-1R) inhibitor enhanced CD8+ T cell motility and infiltration into tumors, and further combination therapy with anti-PD-1 increased the contact between tumor cells and CD8+ T cells [69]. Our previous study suggested that combination therapy using melittin-dKLA and an anti-PD-L1 antibody induces CD8+ T cell activation and depletes M2-like TAMs in a breast cancer model [70]. Therefore, additional studies are needed to evaluate a possible combination therapy using PEG-melittin-dKLA 8-26 and immune checkpoint inhibitors.

## 4. Materials and Methods

### 4.1. Peptide Synthesis

All peptides were purchased from Genscript Corporation (Piscataway, NJ, USA). Fluorescein isothiocyanate (FITC)-Ahx was linked by an amide bond at the N-terminal of the peptides. All peptides were purified to greater than 95% purity. All peptides used in all experiments are listed below: FITC-melittin-random (LSRQRALQIKGLIKTPWKIAGVLGVT), FITC-melittin (GIGAVLKVLTTGLPALISWIKRKRQQ), FITC-melittin 1-14 (GIGAVLKVLTTGLP), FITC-melittin 1-20 (GIGAVLKVLTTGLPALISWI), FITC-melittin 8-20 (VLTTGLPALISWI), FITC-melittin 8-22 (VLTTGLPALISWIKR), FITC-melittin 8-26 (VLTTGLPALISWIKRKRQQ), FITC-melittin 10-26 (TTGLPALISWIKRKRQQ), FITC-melittin 12-26 (GLPALISWIKRKRQQ), FITC-melittin 15-26 (ALISWIKRKRQQ), melittin-dKLA (GIGAVLKVLTTGLPALISWIKRKRQQGGGGS-d[KLAKLAKKLAKLAK]), melittin-dKLA 8-26 (VLTTGLPALISWIKRKRQQGGGGS-d[KLAKLAKKLAKLAK]), and PEG-melittin-dKLA 8-26 ({PEG2}VLTTGLPALISWIKRKRQQGGGGS-d[KLAKLAKKLAKLAK]). The peptides were dissolved in sterile distilled water containing 0.1% acetic acid and diluted in DPBS (Welgene, Gyeongbuk, Republic of Korea).

### 4.2. Cells and Mice

Wild-type BALB/c mice were purchased from DBL (Chungbuk, Republic of Korea). Animal procedures were approved by the University of Kyung Hee Institutional Animal Care and Usage Committee (KHUASP(SE)-20-398). All animals were maintained in a pathogen-free environment with a 12 h light/dark cycle and were supplied with water and food. The 4T1 triple-negative breast cancer (TNBC) cell line was cultured in RPMI1640 media containing 10% fetal bovine serum (FBS; Welgene, Gyeongsan, Republic of Korea), 100 U/mL penicillin, and 100 μg/mL streptomycin (Gibco, Thermo Fisher Scientific Inc., Waltham, MA, USA). The human leukemia monocytic cell line (THP-1) was maintained in RPMI1640 (Welgene) containing 10% fetal bovine serum (FBS; Welgene, Gyeongsan, Republic of Korea), 100 U/mL penicillin, and 100 μg/mL streptomycin (Gibco). The cells were cultured every 2–3 days until 80% confluence, and the cells were incubated at 37 °C in a humidified 5% CO_2_ incubator for all studies.

### 4.3. Macrophage Polarization

THP-1 cells were differentiated into macrophages by incubation with 100 nM phorbol-12-myristate-13-acetate (PMA; Sigma-Aldrich, St. Louis, MO, USA) for 24 h. The cells were incubated with RPMI1640 containing 5% FBS for 48 h for non-polarized M0 cells. For inducing polarization into M1 macrophages, cells were treated with 20 ng/mL recombinant human interferon (rhIFN)-γ (Prospec, Ness Ziona, Israel) and 100 ng/mL lipopolysaccharide (LPS; Sigma-Aldrich) for 48 h. To obtain M2 macrophages, the cells were treated with 20 ng/mL recombinant human interleukin (rhIL)-4 (Prospec) and 20 ng/mL rh IL-13 (Prospec) for 48 h.

### 4.4. Flow Cytometry

To identify each macrophage phenotype, THP-1 cells were seeded at a density of 8 × 10^5^ cells/well in a 6-well plate and polarized into M0, M1, and M2 macrophages using the aforementioned methods. The cells were washed with phosphate-buffered saline (PBS) and harvested using trypsin-EDTA. The cells were washed using wash buffer (BD Biosciences, San Jose, CA, USA) and stained with antibodies. The following antibodies were purchased from BD Bioscience: human CD86-PE-Cy7 and CD163-APC.

To test peptide affinity, THP-1 cells were seeded at a density of 5 × 10^6^ cells/well in 6-well plates. The cells were polarized as described above. After incubation, cells were washed with PBS and harvested using trypsin-EDTA. The cells were further washed with a wash buffer (BD Biosciences) and stained with 100 nM FITC-conjugated melittin or melittin fragments (Genscript) for 1 h at 4 °C. The cells were washed three times with the wash buffer. All data were analyzed using the FACSLyric system (BD Biosciences) and FlowJo software (Tree star, San Carlos, CA, USA).

### 4.5. MTS Assay

THP-1 cells were seeded at a density of 2 × 10^4^ cells/well in a 96-well plate and polarized into M0, M1, and M2 macrophages for 24 h in the same RPMI medium composition and cytokine concentration as described for macrophage polarization. The cells were treated with melittin-dKLA or melittin-dKLA 8-26 (Genscript) in serum-free RPMI and incubated at 37 °C for 20 h. After incubation, the medium was removed and cells were incubated with MTS (3-(4,5-dimethylthiazol-2-yl)-5-(3-carboxymethoxyphenyl)-2-(4-sulfophenyl)-2H-tetrazolium) solution (Promega, Madison, WI, USA) diluted 1:5 in RPMI serum-free media for 1 h at 37 °C. The absorbance was measured at 490 nm using a microplate reader (Molecular Devices, San Jose, CA, USA). All data are expressed as the mean ± SEM of two independent experiments.

### 4.6. Stability Test

Melittin-dKLA, melittin-dKLA 8-26, and PEGylated (PEG)-melittin-dKLA 8-26 were purchased from Genscript Corporation. The peptides were supplemented with 5% human AB serum (Sigma-Aldrich) and incubated for 0 h, 24 h, 48 h, and 72 h at 37 °C. The samples were then transferred to a freezer to stop the serum reaction. Peptide samples were boiled for 5 min at 95 °C, and proteins were separated on 15% SDS Tris-Glycine gel. The gel was stained using D-PLUS^TM^ protein gel staining solution (Dongin Biotech, Seoul, Republic of Korea) for 30 min and washed three times with distilled water. Protein bands were visualized using the ImageJ software (NCI, Bethesda, MD, USA).

### 4.7. Hemolysis Assay

Blood was collected from each BALB/c mouse. Red blood cells were isolated from fresh blood samples by centrifugation at 2000× *g* rpm for 10 min at 4 °C. After centrifugation, isolated red blood cells were washed with phosphate-buffered saline (PBS; Welgene) until the supernatant became clear, after which the supernatant was removed, and PBS was added to suspend red blood cells. The red blood cell suspension was aliquoted, and the peptides were incubated with samples at concentrations of 1, 2, 5, 10, 20, and 40 μM for 4 h at 37 °C. 2% Triton X-100 in PBS was used as the positive control, and PBS was used as the negative control. After incubation, the samples were centrifuged at 2000× *g* rpm for 5 min at 4 °C, and 100 μL of the supernatant was collected and transferred to a 96-well plate. The optical density of the samples was measured at 450 nm using a microplate reader (Molecular Devices). The percentage of hemolysis was calculated using the following equation: hemolysis (%) = (OD_450 nm_ sample − OD_450 nm_ negative control)/(OD_450 nm_ positive control − OD_450 nm_ negative control) × 100%. All data are expressed as the mean ± SEM of three independent experiments.

### 4.8. Animal Study

For the mouse tumor model, 4T1 cells were mixed with Matrigel (Corning, NY, USA). Female BALB/c wild-type mice (6–8 weeks old) were inoculated with 1 × 10^5^ cells per mouse in the left 4th mammary gland (N = 18 per group). Five days after inoculation, melittin-dKLA (Genscript) or PEG-melittin-dKLA 8-26 (Genscript) (200 nmol/kg) was administered intraperitoneally every 3 days, and tumor size and body weight were measured every 3 days using a digital caliper and a digital scale, respectively. The tumor volume was determined using the following equation: (width × width × length)/2. Mice were euthanized after eight injections were administered. For the survival analysis, the experiment was conducted as previously described (N = 15 per group), and mice that died naturally were confirmed. When the tumor volume of the mouse reached a maximum diameter of 2 cm, the mouse was euthanized according to the guidelines.

### 4.9. Quantitative Real-Time PCR

THP-1 cells were differentiated and polarized using the methods described above. Tumor tissues were harvested from mice. Total RNA was extracted from polarized macrophages or tumor tissues using an easy-BLUE RNA extraction kit (iNtRON Biotechnology, Seongnam, Republic of Korea). cDNA was synthesized using Cyclescript reverse transcriptase (Bioneer, Daejeon, Republic of Korea), following the manufacturer’s instructions. Real-time PCR was performed using a CFX connect real-time PCR system (Bio-Rad Laboratories, Hercules, CA, USA) and the SensiFAST SYBR no-Rox kit (Bioline, London, UK). The cDNA synthesis conditions were as follows: for macrophage phenotype identification, cycling conditions were 95 °C for 10 s, 60 °C for 10 s, and 72 °C for 30 s; for confirmation of macrophage marker expression in the tumor, cycling conditions were 95 °C for 10 s and 55 °C for 30 s. The base sequences of the primers used were as follows: human primers: CCR7: forward, 5′-GAT GCGATG CTC TCT CAT CA-3′; reverse, 5′-TGT AGG GCA GCT GGA AGA CT-3′. TNF-α: forward, 5′-GCC CAG GCA GTC AGA TCA TCT-3′; reverse, 5′-TTG AGG GTT TGC TAC AAC ATG G-3′. Arginase-1: forward, 5′-GGC TGG TCT GCT TGA GAA AC-3′; reverse, 5′-CTT TTC CCA CAG ACC TTG GA-3′. CD206: forward, 5′-AAC AGT CAG TCA AGC CCA GG-3′; reverse, 5′-AGG ACA GAC CAG TAC AAT TCA G-3′. β-actin: forward, 5′-CAT GTA CGT TGC TAT CCA GG3-3′; reverse, 5′-CTC CTT AAT GTC ACG CAC GAT-3′), and mouse primers: TGF-β: forward, 5′-CCA CCT GCA AGA CCA TCG AC-3′; reverse, 5′-CTG GCG AGC CTT AGT TTG GAC-3′. Arginase-1: forward, 5′-AGA CAG AGG AGT GAA GAG-3′; reverse, 5′-CGA AGC AAG CCA AGG TTA AAG C-3′. MMP9: forward, 5′-TGA ATC AGC TGG CTT TTG TG-3′; reverse, 5′-ACC TTC CAG TAG GGG CAA CT-3′. β-actin: forward, 5′-GTG CTA TGT TGC TCT AGA CTT CG-3′; reverse, 5′-ATG CCA CAG GAT TCC ATA CC-3′.

### 4.10. H&E Staining

Mouse lungs were fixed in 10% formalin. Lung tissue was embedded in paraffin and sectioned at 5 μm thickness. Tissue sections were deparaffinized with xylene for 10 min. The lung sections were dehydrated and stained with hematoxylin (Cancer Diagnostics, Inc., Durham, NC, USA) for 30 s and washed with tap water for 10 min. After washing, the slides were stained with eosin approximately three times and washed again in tap water for 10 min, and dehydrated with increasing concentrations of ethanol. Tissue slides were mounted and photographed at 1.25× under a light microscope (Olympus, Tokyo, Japan). Lung nodule area and lung nodule counts were calculated using ImageJ software.

### 4.11. Statistics

All data are expressed as the mean ± standard error of the mean. Statistical significance was analyzed using one-way ANOVA followed by Tukey’s multiple comparison tests, two-way repeated measures ANOVA, or Student’s unpaired T test using Prism 5.01 software. Differences were considered statistically significant at *p* < 0.05.

## 5. Conclusions

In conclusion, our findings showed that PEG-melittin-dKLA 8-26 modified from melittin-dKLA was effective in many ways, such as an improving survival, reducing M2-like TAMs, and inhibiting lung metastasis, with minimal side effects. This suggests that PEG-melittin-dKLA 8-26 modified from melittin-dKLA has the potential to replace melittin-dKLA as an antitumor agent.

## Figures and Tables

**Figure 1 ijms-23-15751-f001:**
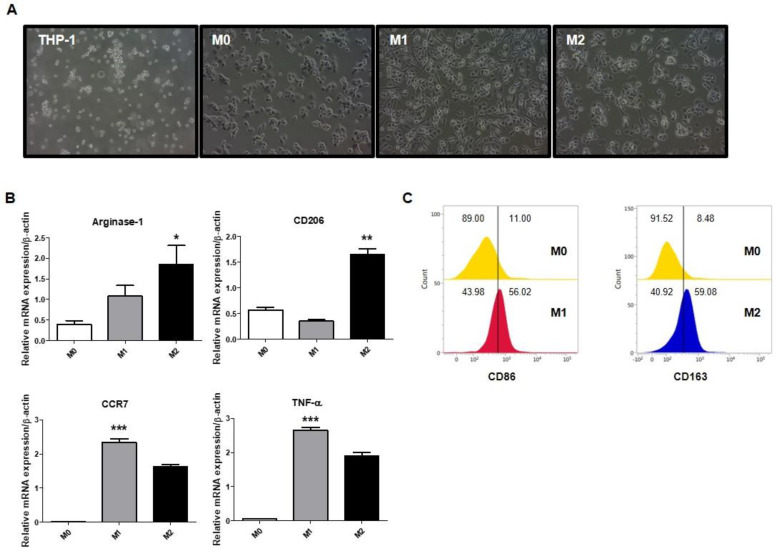
Polarization of macrophages in THP-1-derived macrophages. Human monocyte THP-1 cells were stimulated with 100 nM PMA for 24 h and were polarized into M1 macrophages using 100 ng/mL LPS and 20 ng/mL IFN-γ and into M2 macrophages using 20 ng/mL IL-4 and 20 ng/mL IL-13. (**A**) Morphologies of THP-1 and differentiated M0, M1, and M2 macrophages were examined by light microscopy. Magnification, 20×. (**B**) Expression of Arginase-1 and CD206 as M2 macrophage markers, and CCR7 and TNF-α as M1 macrophage markers was measured by RT-PCR. (**C**) Expression of CD86 and CD163 was detected using flow cytometry. All data are presented as the mean ± SEM; * *p* < 0.05, ** *p* < 0.01, *** *p* < 0.001 versus M0 (n = 3).

**Figure 2 ijms-23-15751-f002:**
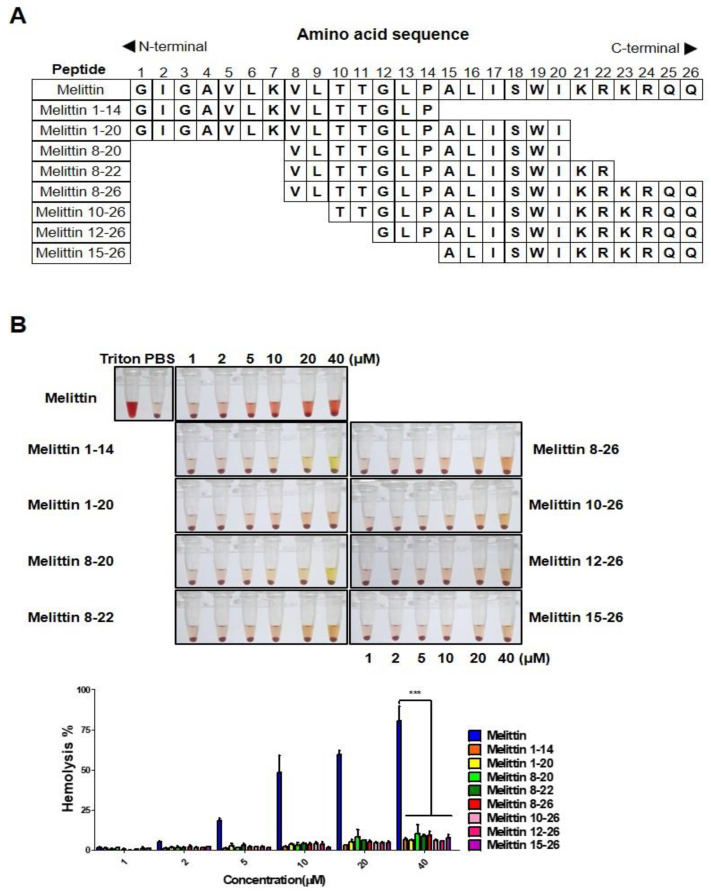
Melittin fragments have fewer hemolytic effects than melittin. (**A**) Amino acid sequence of melittin and melittin-truncated peptide. (**B**) Hemolysis of red blood cells treated with FITC-conjugated melittin or FITC-conjugated melittin fragments at concentrations ranging from 1 μM to 40 μM and incubated at 37 °C for 4 h. 2% Triton-X 100-treated cells were used as the positive control, and PBS-treated cells were used to as the negative control. The optical density of hemoglobin released supernatant was measured at 450 nm. All data are presented as the mean ± SEM; *** *p* < 0.001 versus melittin (n = 3).

**Figure 3 ijms-23-15751-f003:**
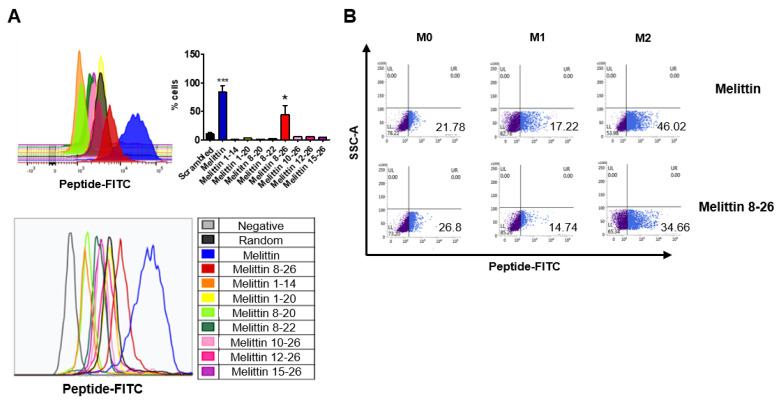
Binding of melittin fragments to the THP-1-derived M2 macrophages. Human monocyte THP-1 cells were stimulated with 100 nM PMA for 24 h and polarized into M1 macrophages using 100 ng/mL LPS and 20 ng/mL IFN-γ and into M2 macrophages using 20 ng/mL IL-4 and 20 ng/mL IL-13. Cells were incubated with FITC-conjugated melittin or FITC-conjugated melittin fragments (100 nM) at 37 °C for 1 h. (**A**) Melittin or melittin fragment affinity for M2 macrophages was measured using flow cytometry. All data are presented as the mean ± SEM; * *p* < 0.05, *** *p* < 0.001 versus the scrambled peptide (n = 3). (**B**) The affinity of melittin or melittin 8-26 for macrophages was detected by flow cytometry.

**Figure 4 ijms-23-15751-f004:**
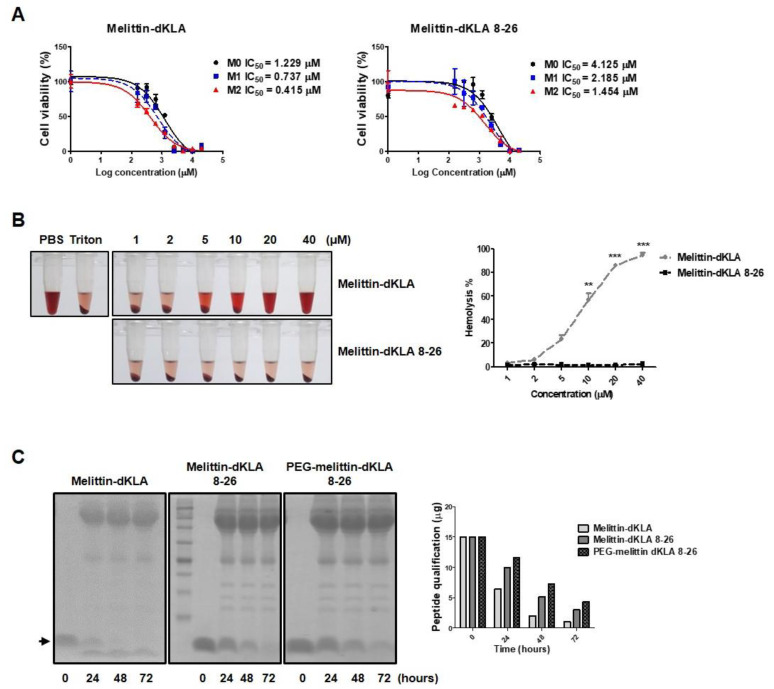
Characterization of the optimized hybrid peptide drug, PEG-melittin-dKLA 8-26. (**A**) MTS assay of melittin-dKLA or melittin-dKLA 8-26 in macrophages. Human THP-1 was stimulated with 100 nM PMA for 24 h and polarized into M1 macrophages using 100 ng/mL LPS and 20 ng/mL IFN-γ and into M2 macrophages using 20 ng/mL IL-4 and 20 ng/mL IL-13. Cells were treated with melittin or melittin-dKLA 8-26 at concentrations ranging from 0 to 20 μM for 20 h. After incubation, the media were removed, and cells were treated with MTS (3-(4,5-dimethylthiazol-2-yl)-5-(3-carboxymethoxyphenyl)-2-(4-sulfophenyl)-2H-tetrazolium) solution diluted in serum-free RPMI medium for 1 h. The optical density was measured at 490 nm. All experiments were performed in duplicates. (**B**) Hemolysis of melittin-dKLA or melittin-dKLA 8-26. Melittin-dKLA or melittin-dKLA 8-26 was added to red blood cells at concentrations ranging from 1 μM to 40 μM and incubated at 37 °C for 4 h. 2% Triton-X 100-treated cells were used as the positive control, and PBS-treated cells were used as the negative control. The optical density of hemoglobin released in the supernatant was measured at 450 nm. The data are presented as the mean ± SEM; ** *p* < 0.01, *** *p* < 0.001 versus melittin-dKLA (n = 3). (**C**) Melittin-dKLA, melittin-dKLA 8-26, or PEG-melittin-dKLA 8-26 stability was examined via a stability test. Human serum was added to 15 μg of peptides and incubated at 37 °C for 0 h, 24 h, 48 h, and 72 h. Protein bands were visualized and measured by Image J.

**Figure 5 ijms-23-15751-f005:**
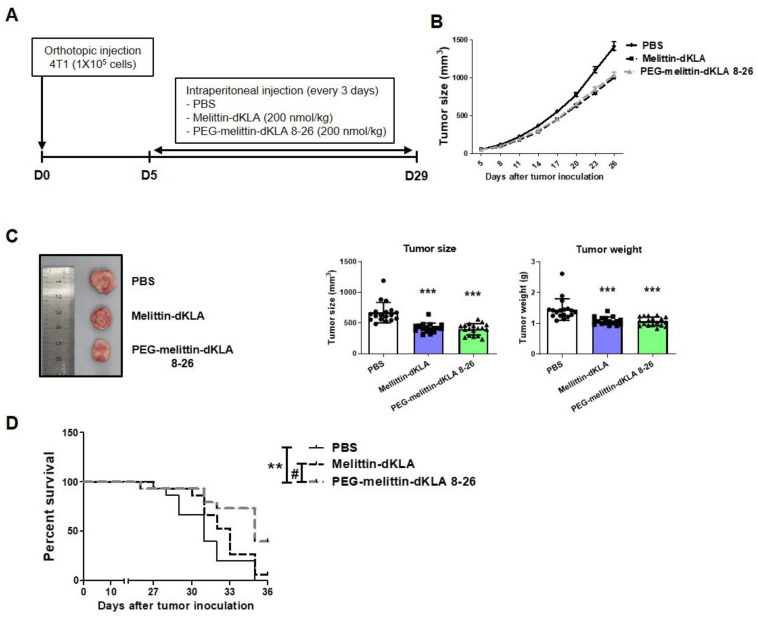
Inhibition of tumor growth and increased survival rates by PEG-melittin-dKLA 8-26 in the 4T1 TNBC mouse model. (**A**) In vivo experiment schedule. (**B**) The volume of the tumor was measured using a digital caliper. Tumor volume was calculated using the following equation: (width × width × length)/2. (**C**) To access tumor volume and weight, mice were euthanized on day 29 after tumor inoculation. Tumor tissues were imaged, and tumor volume and weight were measured using a digital caliper and electronic scale, respectively, after sacrifice. All data are presented as the mean ± SEM; *** *p* < 0.001 versus the PBS group (n = 18). (**D**) Mouse survival was calculated as the life span from the day of tumor inoculation. The median survival and *p* values were determined using the Log-rank test. ** *p* < 0.01 versus the PBS group, ^#^ *p* < 0.05 versus the melittin-dKLA group (n = 15).

**Figure 6 ijms-23-15751-f006:**
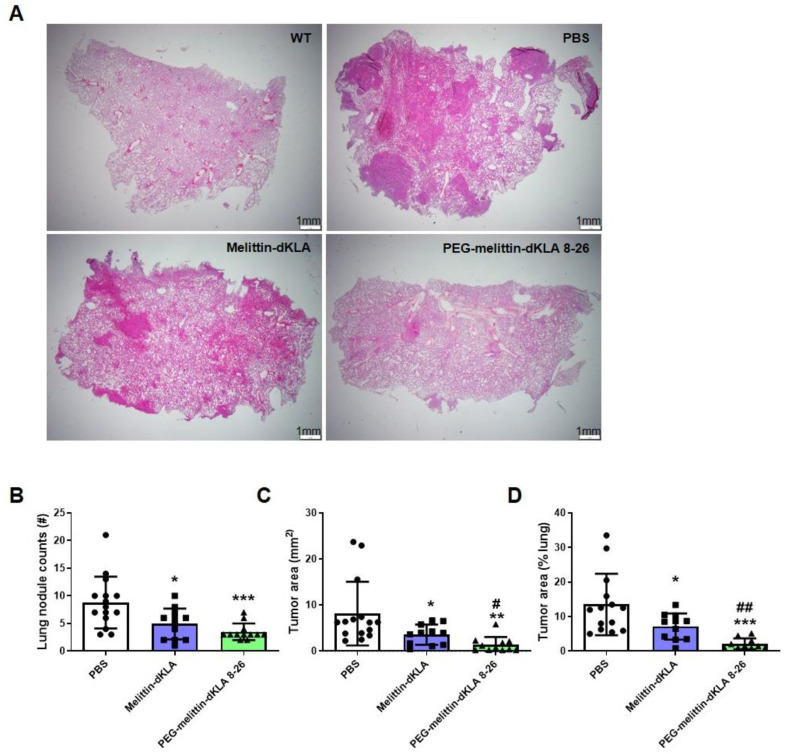
Enhanced suppression of lung metastasis by PEG-melittin-dKLA 8-26 in the TNBC mouse model. (**A**) Image of H&E-stained lung tissue. Magnification, 1.25×. Scale bar, 1 mm. (**B**) Lung nodule counts and (**C**,**D**) lung nodule area were calculated using Image J. All data are presented as the mean ± SEM; * *p* < 0.05, ** *p* < 0.01, *** *p* < 0.001 versus the PBS group, ^#^ *p* < 0.05, ^##^ *p* < 0.01 versus the Melittin-dKLA group (n = 12).

**Figure 7 ijms-23-15751-f007:**
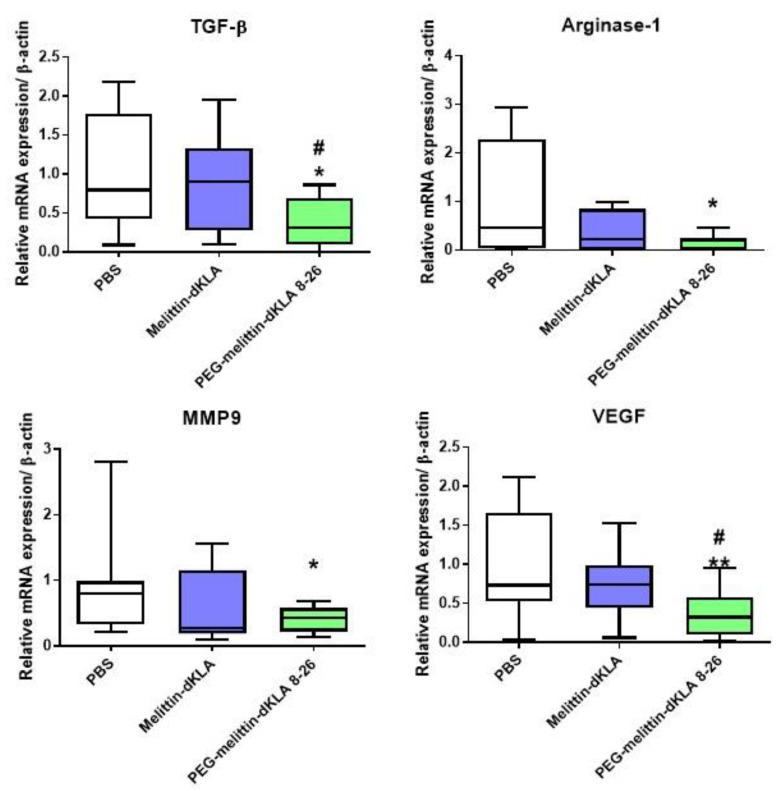
Reduction in M2-like TAMs by PEG-melittin-dKLA 8-26 in the 4T1 TNBC mouse model. Mice were euthanized on day 29 after tumor inoculation, and mouse tumor tissues were harvested. RNA was extracted from tumor tissues. Expression of TGF-β, arginase 1, MMP9, and VEGF was measured using RT-PCR. All data are presented as the mean ± SEM; * *p* < 0.05, ** *p* < 0.01 versus the PBS group; # *p* < 0.05 versus the melittin-dKLA group (n = 10–13).

## Data Availability

All data generated or analyzed during this study are included in this published article.

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
