# Peer review of "Enhanced Therapeutic Effect of Optimized Melittin-dKLA, a Peptide Agent Targeting M2-like Tumor-Associated Macrophages in Triple-Negative Breast Cancer"

_ijms, 2022, doi:10.3390/ijms232415751_

Round 1

Reviewer 1 Report

I thank the authors very much for the submission of their manuscript. The manuscript is well written but some issues need to be addressed before its publication.

Minor issues

1. Figure 5. Meaning of p# must be indicated in the figure legend.

2. Line 175-175: The authors mention a lot of details about the concentration of treatment reagents and tumor cells (e.g. line 175-178). This is not necessary. It is sufficient when this data is included in the M&M section (applies to the entire manuscript).

3. Figure legends: P as a measure of significance is generally written in lower case.

4. Minor spelling mistakes:

e.g. Line 92/93/101/153(grammar)/line 158  (should be “human THP-1 cells” instead of “Human monocytes THP-1”)

e.g. Line 162-163: Should be rephrased as: “After incubation media was removed and cells were treated with MTS solution diluted in serum-free RPMI 162 medium for 1h.”

5. Some abbreviations are missing: e.g. MTS

6. The studied compound is written as (PEG)-melittin-dKLA 8-26 in the text but as PEGMelittin-dKLA 8-26 in Figure 5. This should be coherent throughout the manuscript.

7. Line 263-278 This information does not belong into the discussion section and should be moved into the introduction when melittin and KLA are mentioned.

Major issues

1. Fig.1/2: Although the data looks convincing, it is questionable to perform statistical analysis or to draw conclusions from experiments that were only performed in duplicates (n=2). In general, the exact value of n should be indicated in each figure legend.

2. Fig. 5 C. The time point when tumors were analyzed must be indicated in the legend.

3. It is puzzling that PEG-melittin-dKLA 8-26 treatment does not provide any benefit on tumor growth and tumor size but on overall survival. Is this solely due to the inhibitory effect on metastasis formation? This should be mentioned in the discussion

4. Fig.7 It should be clarified from where the RNA samples were obtained (primary tumor/metastasis) and at which time point.

5. The authors claim that the melittin-dKLA 8-26 specifically targets M2 macrophages. The factors that the authors analyzed to support their hypotheses can also be produced by other cell types like neutrophils, PMN-MDSC or endothelial cells. Based on the data presented, it is not possible to directly link the observed survival benefit to a specific effect of melittin on M2 macrophages in vivo. The manuscript has to be revised accordingly.

6. Fig. 3A How does melittin discriminate between different cell types. Is anything known about potential receptors for melittin? I assume it is not associated with the phagocytic activity of macrophages since M1 cells are generally considered to have a higher phagocytic potential. The authors describe that melittin binds to cell membranes based on hydrophobic interactions, but why is the effect stronger for M2 macrophages? This should be adressed in the discussion.

7. Fig. 7B It should be indicated how many samples were measured. From the figure I assume that the data was analyzed in triplicates. The a-tubulin band of the last replicate of the melittin-dKLA and the melittin-dKLA 8-26 sample indicate unequal loading of the gel. Therefor, this data cannot be used and should be removed from both the gel blot and the quantification on the right. Also, it should be indicated how the bands on the western blots were quantified

I hope that these comments are of help to the authors and I am looking forward to a revised version of the manuscript.

Best regards,

Dr. Christopher Groth

Reviewer 2 Report

The authors have developed an optimized melittin-dKLA targeting M2-like tumor-associated macrophages. Some new peptide agents have shown a better hemolytic effect but similar binding ability. In vivo studies also suggested that tumor-growth inhibition was also similar between melittin and optimized melittin. It is a nice paper with well-designed experiments and meaningful discussions. I agree to have it published on IJMS once some revision is finished. 

1. Page 2, Line 71 - 73: Please further explain IFN-γ, IL-13, and IL-4. 

2. Do you have any characterization data for the peptides? 

3. Could you have a separate conclusion section? 
